# Manipulating risk of infection and appeal to public benefit increase compliance with infection control measures in a hypothetical pandemic scenario

**Sebastian Bjørkheim** *, **Bjørn Sætrevik**

Department for Psychosocial Science, Operational Psychology Research Group, Faculty of Psychology, University of Bergen, Bergen, Norway

* sebastian.bjorkheim@uib.no

## Abstract

To limit an infectious outbreak, the public must be informed about the infection risk and be motivated to comply with infection control measures. Perceiving a situation as threatening and seeing benefits to complying may be necessary to motivate for compliance. The current study used a preregistered survey experiment with a 2-by-2 between-subject design to investigate if emphasizing high infection risk and appealing to societal benefits impacted intention to comply with infection control measures. The results from a representative Norwegian sample (N = 2533) show that describing a high (as opposed to low) personal risk scenario had a small main effect on compliance. Further, appealing to public (as opposed to self-interested) benefits also had a small main effect. There was no interaction between risk scenario and motivational emphasis. The results suggest that to maximize compliance, information about disease outbreak should emphasize the individual risk of contracting the disease, and could also underline the public value of limiting infection spread. These findings can inform health authorities about the motives underlying compliance with infection control measures during an infectious disease outbreak.

## Introduction

Infectious outbreaks like the COVID-19 pandemic posed a serious threat to global health and an exceptional challenge to health authorities. How people perceive the pandemic risk and the extent to which they are motivated to comply with infection control measures may determine the societal impact of the pandemic. Health authorities are typically granted the responsibility to inform about health risks and encourage the public to take preventive actions. Compliance with infection control measures is partly determined by psychological factors such as perceived risk, risk literacy, self-efficacy, and the perceived benefit and cost of complying. These factors may be influenced by how risks and benefits are presented in a given situation.

**Data Availability Statement:** All relevant data are linked to within the paper and can be found at (https://osf.io/6vwh4/).

**Funding:** The present study was conducted as a part of the PANDRISK research project funded by the Trond Mohn Foundation, project number TMS2020TMT08. The funder had no role in study design, data collection and analysis, decision to publish, or preparation of the manuscript.

**Competing interests:** The authors have declared that no competing interests exist.

## Perceived risk

Risk is typically considered to consist of the probability (or likelihood) of a negative event factored by the severity of the outcome. Perceived risk is the subjective assessment of the two components [1]. Although the probability of an event and the expected outcome can be established statistically in some cases, perceived risk has been shown to deviate from objective measures of risk [2, 3]. The individual perception of risk may depend on statistical literacy and the way in which risk is communicated [4]. Information about personal risk may thus be overestimated or underestimated as a consequence of how public authorities communicate a risk scenario. While the optimal method for conveying health risks has yet to be agreed upon, it has been found that visual aids and information about absolute event rates improves risk understanding among patients [5].

The "protection motivation theory" proposes that perceived severity of threat, vulnerability, self-efficacy and response effectiveness facilitates the adaptation of preventive behaviours [6, 7]. During a pandemic, threat severity and threat vulnerability may correspond to how people asses the likelihood of being infected and how severe such an event would be. Previous research [8–10] has pointed to perceived risk as a key predictor of compliance with infection control measures. For example, during the H1N1 pandemic in the Netherlands, Bults and colleagues [11] found that people who considered the risk of infection to be high were more willing to comply with government advice. Similarly, a longitudinal study in Germany [12] found that low risk perception were among the main contributors to vaccine avoidance. A review of the attitudinal and demographic determinants of compliance found that older, female and participants with higher education that viewed the disease as threatening were more likely to engage in protective behaviours [13].

Recent work on the COVID-19 pandemic has also pointed to perceived risk as an important factor for explaining compliance with infection control measures. A survey of Americans during the early phase of the pandemic (March 11–16) found that proclivity to follow infection control measures was predicted by the perceived severity of contracting COVID-19 [14]. The association between perceived risk and knowledge of COVID-19, and the adoption of protective measures has been found in a number of countries [e.g. in China; 15; in Saudi Arabia; 16; Egypt and Nigeria; 17; and in Germany; 18]. In addition, a literature review of adherence to quarantine measures found that perceived risk, knowledge about protocols and seeing the measure as valuable were associated with compliance [19]. While the literature on health protective behaviour often emphasizes personal risk as a motivating factor for precautionary behaviour, people may also be motivated to protect others or society at large. If feeling at risk is essential for compliance, information that emphasize personal and societal risks may increase compliance with infection control measures.

## Motivation

People may have different motivations for complying with infection control measures. The "health belief model" posits that people's engagement in precautionary behaviour is based on deliberative weighing of costs and benefits of such behaviour [20]. Compliance often implies a cost for the individual, as it may include avoiding desirable actions such as travel and social gatherings, and performing undesirable actions such as wearing facemask in public or working from home. Benefits to compliance are on the other hand shared by everyone, by mitigating the societal impact of the disease. As such, wearing a facemask or keeping physical distance to others may be perceived as a prosocial act, as one incurs inconvenience to oneself for the benefit of the community [21]. Research on health protective behaviour has typically assumed that people are interested in protecting themselves from health risks. But recently it has also been

emphasised how people are motivated to protect the welfare of others [22, 23]. Moreover, a survey from the first weeks of the COVID-19 outbreak in Norway found that prosocial behavioural intentions were prevalent and that most people intended to comply with infection control measures [24].

When health authorities encourage people to take precautions, they may emphasize personal benefits such as decreasing the risk of infection and becoming sick or prosocial benefits such as easing the burden on the health system. A salient example of the latter is the appeal to help "flatten the curve" of COVID-19 spread during the early phases of the pandemic [25]. Flattening the curve was in part promoted to spare the hospitals from overload, and may be viewed as a prosocial encouragement for compliance with infection control measures. An experiment on vaccine uptake showed that information about both personal benefits and collective benefits increased willingness of vaccination compared to a control vignette [26]. However, the effect was stronger for the collective benefits treatment where information about herd immunity was emphasized. The efficacy of motivational emphasis may also be influenced by perceived vulnerability towards a disease. A recent study on influenza vaccine uptake found that an emphasis on social benefit induced vaccination among people who perceived themselves to be at low risk, while people who self-categorize as at-risk were more swayed by appeals to personal benefit [27]. This suggest that the motivation for compliance may interact with people's sense of being vulnerable and that health communication can benefit from tailoring information to different risk-groups. However tailoring health communication to persuade people into adopting health protective behaviour may also have downsides, such as reducing the credibility of the recommendation and reducing trust in the messenger [28]. Moreover, emphasizing arguments in favour of taking precautions has also been found to be ineffective in increasing vaccine uptake [29].

Care for the welfare of other people have recently been found to predict compliance with infection control measures during the COVID-19 pandemic [30]. Furthermore, they found that compliance was more strongly associated with perceived risk for the public than with personal risk. An experiment on the effect of motivational frames to induce protective behavior [31], found that messages that emphasized public rather than personal benefits fostered greater intentions to comply with infection control advice in the early stages of the pandemic. Yet, in subsequent experiments conducted later in the pandemic they found that the frames were similarly effective in fostering preventive intentions. A non-representative online study conducted in the initial weeks of the COVID-19 pandemic found that self-interested frames ("protect yourself") were more effective than distant prosocial frames ("protect your community") in eliciting clickthrough's to Centers for Disease Control recommendations [32]. While this finding supports the efficacy of self-interested frames over prosocial frames, they also found that a more personal prosocial framing ("protecting loved ones") were similarly effective as the self-interested message. These studies indicate that appeals to both personal benefit and collective benefit can induce protective behaviour, but their relative importance is less well understood.

## Research needs

To handle a disease outbreak, public authorities should inform and encourage behaviour that limits the infection spread. Research has suggested that perceived personal risk and prosocial motivation may both be predictors of compliance to infection control measures. This literature is typically correlational, and as such may be limited by "third variable problems", for example that personality causes associations between risk perception, motivation and compliance. Another limitation for correlational studies is the "directionality problem", for example that people who for some reason see themselves as less compliant also deemphasize the risk in order to justify their actions.

How people perceive risk is related to their ability to control their level of exposure and it may thus be difficult to determine the impact of perceived risk on compliance from correlational data alone [33]. In addition, previous studies have suggested that both self-interest and prosocial motives induce protective behaviour, but the relative importance of each is less known. It is therefore of interest to investigate whether compliance is caused by level of infection risk and to discover the relative importance of self-interest and prosocial motives. Since infection risk and benefit for compliance coincide during an infectious outbreak, it is also of interest to investigate if level of risk interacts with motivational emphasis to comply with infection control measures. Knowledge about the relative importance of self-interested and prosocial motives in different risk scenarios can inform health authorities about how people respond to infection control measures during different phases of disease outbreak.

## Current study

The current study is part of a research project which aims to measure, track and predict the effect of perceived risk on compliance during the COVID-19 pandemic in Norway. The project also examines a number of other aspects related to the pandemic throughout the pandemic trajectory in Norway (see website: https://www.uib.no/en/pandrisk). The current study uses data from the fourth nationally representative survey data collection of this project.

**Aims.**   Earlier findings have indicated that perceived risk, and the desire to protect oneself and others may be important predictors of compliance to infection control measures. The aim of the current study is to test causal relationships between risk framing and type of motivation on compliance with infection control measures, and to examine if these factors interact.

**Approach.**   The study fielded a survey experiment to a large nationally representative sample. The experiment was devised as a 2-by-2 between-subjects design with experiment conditions that varied according to the risk of infection (high-risk or low-risk) and benefit for complying (prosocial or self-interest).

**Hypotheses.**   The hypotheses were preregistered on 14th of December 2020, before data was available to the researchers (https://osf.io/ahfdn). Based on the expected association between concern for others and compliance with infection control measures, we registered (H1) *"We expect a main effect for "prosocial motivation" (in both the "high" and "low" risk scenario), such that participants should indicate greater compliance with the advice when the message emphasizes the public benefits of following the advice."* Based on the existing research pointing to perceived risk as a possible predictor of compliance, we registered (H2) *"We expect a main effect for "high-risk" scenario, such that participants should indicate greater compliance with the advice when the description emphasizes the high level of contagion in the scenario".* Finally, since it is possible that self-interested motivation may be more important in situations with greater personal risk while prosocial motivation is more important in low-risk scenarios, we also expected an interaction effect of risk and motivation such that (H3) *"We expect an interaction effect so that "self-interested motivation" has a larger effect in the "high-risk" scenario".* "Self-interested motivation" was phrased as "egocentric motivation" in the preregistration. The threshold of significance was set at .05.

In addition to these preregistered hypotheses, we also wanted to explore whether the response to the survey experiment was influenced by the extent of risk experienced by the pandemic that was ongoing during the data collection.

## Methods and materials

### Participants

The data was collected using the "Norwegian Citizen Panel", which is a running, online survey of Norwegians' opinions towards important social matters. The panel has been fielded three to

four times a year since 2013 and contain questions about diverse social matters. Researchers at the University of Bergen are responsible for running the panel, while the company Ideas2Evidence manages the recruitment of participants and production of survey.

Compared to a perfectly representative sample of the Norwegian population, there is some underrepresentation in terms of age, education level, and geographical residency. People over the age of 60 are overrepresented by 16% while those under 29 years old are underrepresented by 13%. Those with a university or college degree are overrepresented by 29%, while those with upper secondary education are underrepresented by 10% and those with elementary education are underrepresented 19%. In regard to geography, the sample reveal a slight overrepresentation of people who resided in Oslo and Akershus (5%) and Western Norway (2%), while people from Eastern and Northern Norway were underrepresented by 4% and 2% respectively. These deviations from representativity are similar to previous samples collected by the panel (see methodology report: https://osf.io/b9nh2/), and are thus unlikely to be related to the topic of the current survey. The dataset is provided with weighting variables to adjust for the deviations in representativity.

In the current data collection, invitations were sent to out to 16.212 eligible respondents of which $N$ = 12.460 filled out the questionnaire (76.8% response rate). The sample was randomly split into six sub-samples, yielding a sample size of 2.533 participants that participated in the current study. This was a larger sample size than we anticipated when preregistering the study. Females constituted 50.2% of the sample (n = 1.272). Almost half (48.1%) of the participants were born in 1959 or earlier, 45.6% were born between 1960 and 1989, and 6.3% were born in 1990 or later. Most (63%) had completed university or college education, while 30% had completed upper secondary education and 4.8% had completed elementary school or had no formal education.

## Data collection procedure

Initial recruitment to the Norwegian Citizen Panel was based on random selection from the Norwegian Tax Administration registry. All Norwegian citizens above the age of 18 are eligible to be recruited and the panel aims to be representative for the adult Norwegian population across several demographic variables. Participants were invited to the current data collection by email sent out on the 2nd of November 2020, with a reminder being sent out a week later to those who had not opened or completed the survey. Subsequent reminders were sent by email and SMS to panel members three times before the 19th of November, but most of the responders had completed the survey within the first few days (see methodology report: https://osf.io/b9nh2/).

The survey was fielded between the 2nd and 27th of November 2020. At this time there was an upsurge in coronavirus cases in Norway. The number of recorded cases rose from 1 540 cases on average per week in October 2020 to 3 794 cases on average per week in November 2020. This period has been referred to as the "second wave" of infection spread and a number of infection control measures were in place at the time [34]. At this time the current recommendations from The Norwegian health authorities were to practice social distancing, to avoid domestic and international travel, comply with quarantine and isolation requirements and take hygiene precautions. In addition, the government restricted organized sport and other recreational activities such as eating in restaurants. The rise in COVID-19 cases and the appropriateness of the infection control measures was prominent in the public debate during the data collection period.

All ethical aspects of the data collection and data storage in the Norwegian Citizen Panel are approved by the Norwegian Centre for Research Data (reference number: 118868). A written informed consent form was obtained from all panel members ahead of the data collection.

The current study did not apply for additional ethical approval as the study procedure did not deviate from the standard practice of the Norwegian Citizen Panel. The study is conducted in accordance with institutional ethical guidelines and does not require individual approval.

## Variables

The experiment was presented as part of a 29-item survey about different aspects of the COVID-19 pandemic (full materials available online at: https://osf.io/mx8gc/). On average, the participants used between 15 and 16 minutes to complete the survey. The experiment described a hypothetical scenario similar to the coronavirus pandemic in Norway but did not mention the COVID-19 pandemic specifically. The participants read an introduction asking them to imagine that the health authorities informed them about the extent of infection risk in their area and among people like themselves, and then appealed to different reasons for complying with infection control measures. The statement describing the extent of infection risk was framed as either high-risk (translated from Norwegian): "*there is a serious outbreak of infection in the area where you live among people in the same life situation as you*" or low-risk: "*only a few people are infected in the area where you live among those in the same life situation as you*". The second statement described different benefits for complying with the infection control measures. This was framed as either personal benefits: "*that you will be safer if you follow all infection control advice*" or prosocial benefits: "*it is important that you follow all infection control advice in order to stop the outbreak*". The difference in the second statement was intended to emphasize the self-interested choice weighting individual benefits (in the first case), as opposed to the importance for societal handling of the pandemic (in the second case).

Since our previous measures in the project had indicated that behavioural intentions for compliance tended to be high [24], we chose to use a more extreme phrasing of our outcome measure, to prevent a ceiling effect. Participants responded to "*How certain are you that you would follow the measures in such a situation?*" on a single Likert-type item with 7 degrees agreement. The scale ranged from 7 = "*Completely certain I would always follow the measures*" to 1 = "*Completely certain that I would not follow the measures*". All analyses were run in Rstudio (see analysis script online at: https://osf.io/mx8gc/).

## Results

### Descriptive statistics for intentions of complying to infection control measures

The response distribution showed that across experiment conditions most participants intended to comply with the infection control measures. As shown in Table 1, the far majority of participants indicated that they were "completely certain" or "quite certain" that they would

**Table 1. Number of responses and percentage share for each response option.**

| How certain are you that you would follow the measures in such a situation? | N | % |
|---|---|---|
| Completely certain that I would always follow the measures | 1073 | 42,4% |
| Quite certain that I would always follow the measures | 1060 | 41,8% |
| I think that I would always follow the measures | 315 | 12,4% |
| I am not sure if I would follow the measures or not | 49 | 1,9% |
| I do not think I would follow the measures | 10 | 0,4% |
| Quite certain I would not follow the measures | 10 | 0,4% |
| Completely certain I would not follow the measures | 8 | 0,3% |
| Did not answer | 8 | 0,3% |

**Table 2. Intention to comply with infection control measures by demographic groups.**

| Demographic group | Intention to comply with infection control measures | | | | | |
|---|---|---|---|---|---|---|
| | *N* | *M* | *SD* | *t* | *F* | *p* |
| Gender | | | | 7.850 | | < .001 |
| Female | 1268 | 6.35 | .82 | | | |
| Male | 1257 | 6.08 | .89 | | | |
| Highest completed education | | | | | .18 | .602 |
| Elementary school | 122 | 6.23 | .90 | | | |
| High school | 766 | 6.22 | .84 | | | |
| University degree | 1590 | 6.21 | .87 | | | |
| Age group | | | | | 13.69 | < .001 |
| 18–30 | 161 | 6.06 | 1.04 | | | |
| 31–60 | 1153 | 6.17 | .89 | | | |
| 61- | 1211 | 6.27 | .81 | | | |
| Gross annual income (USD) | | | | | 5.84 | .016 |
| 30 000 or less | 427 | 6.24 | .92 | | | |
| 31–50 000 | 856 | 6.25 | .79 | | | |
| 51–70 000 | 690 | 6.21 | .86 | | | |
| 70 000 or more | 501 | 6.12 | .91 | | | |

Note: The gross annual income was converted from Norwegian Krone (NOK) to USD at a rate of 1 USD = 9.75 NOK

comply with the measures, while very few were similarly confident that they would not comply. Intention to comply varied somewhat by demographic background (see Table 2). There are too few participants with non-European ethnicity in the sample to investigate whether compliance varies by ethnic background.

## Confirmatory tests of hypotheses

Before performing the hypothesis testing analyses, we verified that the data was suitable for ANOVA testing. The data met the assumption for homogeneity of variance as determined by a Levene's test ($F(3, 2521) = 1.007$, $p = .38$).

**Main effect of high risk.** The preregistered hypothesis H2 anticipated a main effect for "high risk" such that compliance with infection control measures should be greater when the scenario described a situation with a high risk of infection. A two-way ANOVA found a small main effect ($F(1, 2521) = 68.14$, $p < .001$, $\eta2 = .03$) in the predicted direction. According to conventional interpretations, this factor explained a small proportion of the observed variation in compliance.

**Main effect of prosocial motivation.** The preregistered hypothesis H1 anticipated a main effect for "prosocial motivation" such that compliance with infection control measures should be greater when the message emphasized the public benefits of following the advice. A two-way ANOVA found a small main effect ($F(1, 2521) = 7.9$, $p = .004$, $\eta2 = .001$) in the predicted direction. According to conventional interpretations, this factor explained a very small proportion of the observed variation in compliance.

**No interaction between motivation and risk.** The preregistered hypothesis H3 anticipated an interaction effect between "self-interested motivation" and "high risk" such that compliance should be greater in the "high risk" scenario where self-interest was emphasized. A two-way ANOVA failed to find an interaction effect between motivation and risk scenario ($F(1, 2521) = 1.01$, $p = .31$). See Fig 1.

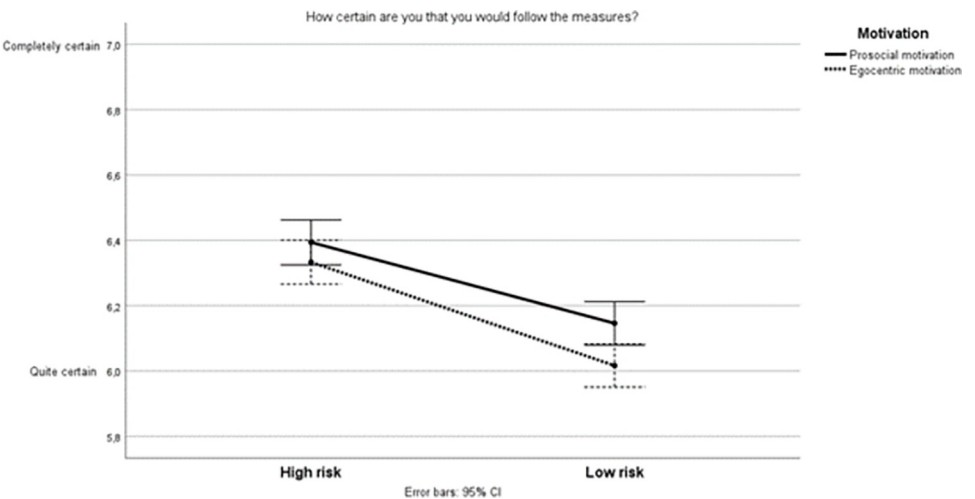

**Fig 1. Main effects for high risk and prosocial motivation on compliance.**

Note that the full scale is truncated from 1–7 to show difference between conditions. A very large share (84,2%) of the responses were between 6 and 7 shown here.

### Explorative analyses

The above analyses are confirmatory tests of the preregistered hypotheses that the framing of a hypothetical future pandemic should influence compliance. The data collection was done in November 2020 during the COVID-19 pandemic, along with measures of attitudes and intentions about the ongoing pandemic. We may use this to explore whether the answers about a hypothetical future pandemic differ depending on the participants' thoughts about the ongoing pandemic. Risk perception and prosocial motivation during the COVID-19 pandemic may have an impact on how participants responded to the hypothetical risk scenario. As they were not preregistered, these analyses should be considered explorative. They are only described briefly here, see more details from these analyses in the online supplemental materials.

**Effect of perceived risk for COVID-19 infection in ongoing pandemic.** The extent to which people perceive the ongoing pandemic as threatening, may have an effect on how they respond to the hypothetical pandemic risk scenario. One item in the survey asked the participants to assess the current risk of being infected with COVID-19 during 2020. The answers were reported on a five-point scale ranging from "very low" to "very high", and overall the participants considered it moderately likely that they would be infected (M = 2.52, SD = .92). This variable was added to the ANOVA described above, resulting in a three-way ANOVA of current risk, hypothetical risk, and motivational emphasis on compliance. The results showed that the current risk did not have a main effect or interaction with the other variables ($p > .18$). This indicates that the current risk does not impact compliance in hypothetical risk scenario. The absence of any interaction effect indicates that the level of hypothetical risk and motivational emphasis impact intentions to comply with infection control measures even after we account for how people perceived the risk in the ongoing pandemic situation. How the risk of COVID-19 infection is perceived thus seem to be of little importance when encountering a hypothetical risk scenario.

**Effect of prosocial motivation for complying in ongoing pandemic.** The extent to which people were pro-socially motivated to comply with the infection control advice in the ongoing pandemic, may have an effect on how they respond to the hypothetical pandemic

scenario. One item in the survey asked the participants if they complied with the current measures in order to "protect others from getting sick". This may be viewed as a prosocial motivation for complying with the measures in the ongoing pandemic. The answers were reported on a five-point scale ranging from "completely disagree" to "completely agree". Overall, the participants reported a very high degree of motivation to protect others (M = 4.52, SD = .63). This item was added to the ANOVA described above, resulting in a three-way ANOVA of hypothetical risk, motivational emphasis and current prosocial motivation on compliance in the hypothetical pandemic scenario. The results showed that prosocial motivation to comply with infection control measures in the ongoing pandemic had a main effect on hypothetical compliance in the experiment (F(1, 2511) = 195.41, p < .001, η2 = 0.07), but there were no interaction effect with either risk framing or motivational emphasis (p > .27). This indicates that people who were motivated to "protect others from getting sick" during the COVID-19 pandemic, also reported higher intentions to comply with infection control measures in a hypothetical infectious disease scenario. The absence of interaction effects indicates that the impact of risk framing and motivational emphasis are not influenced by the level of prosocial motivation during an ongoing pandemic.

## Discussion

### Summary of results

We performed an online survey experiment on a large representative sample of Norwegians during the "second wave" of disease spread of the COVID-19 pandemic in Norway. The experiment varied the extent to which infection risk was high or low, and emphasized either self-interested or prosocial benefits for complying with infection control measures. The results showed a significant but small effect of risk framing and a significant but very small effect of the type of motivational emphasis. Participants who responded to a scenario describing high-risk (i.e., serious outbreak) indicated higher compliance to infection control measures, compared to those who received a low-risk scenario (i.e., relatively few new cases). Further, people who received the scenario appealing to public benefit for compliance (i.e., to stop the outbreak) indicated higher compliance than those who received a personal motive for compliance (i.e., you will be safer). The data showed no interaction between experiment conditions, suggesting that prosocial benefits and infection risk induce compliance independent of each other.

### Possible relationships between risk, prosocial motivation, and compliance

The protection motivation theory [6, 7] posits that perceiving a situation as a health threat facilitates the adoption of preventive measures. Whether people comply with infection control measures may depend on whether they see the infection risk as a threat either for themselves or for society at large. Thus, compliance may vary depending on whether a message emphasizes self-interested or prosocial reasons, or both, for compliance.

We found a small main effect of risk framing, in the sense that across motivational emphasis, intentions to comply was higher in the high-risk scenario. This is in line with previous research as it is typically assumed that perceiving a situation as threatening will motivate people to take precautions, while seeing a situation as safe can induce indifference towards protective behavior [8]. It also supports previous research indicating associations between perceived risk and compliance from the H1N1 pandemic [11, 12] and similar findings from correlational data in the COVID-19 pandemic [14, 19]. Similar to Bish and Michie (2010), we also found that female participants indicated higher intentions to comply than men, and that older (60 or above) participants were slightly more certain of complying than younger participants. An exploratory analysis found that the perceived risk of contracting COVID-19 did not explain

compliance in the experiment. Also, there were no interaction effect between the perceived risk of contracting COVID-19 and main effects of risk framing and motivational emphasis.

We found a small main effect of prosocial motivation, in the sense that across risk framing, intentions to comply was higher when the motivational framing emphasized the societal benefits for compliance. This finding concurs with conceptually similar studies on self-interested versus prosocial framing in the COVID-19 context [23, 31], but runs contrary to an American study who found evidence for the advantage of self-interested framing [32]. It is interesting to note that Jordan and colleagues [31] found prosocial framing to be advantageous in experiments conducted in the early phase of the COVID-19 pandemic (March 14–16, 2020), but did not find the same effect in experiment conducted as the pandemic had progressed (April 17–30, 2020). This may suggest that the underlying motivation to comply with control measures change over the pandemic trajectory or that the effect is particularly sensitive to external circumstances. In this regard, it is worth noting that the current study was conducted during the "second wave" of infection spread in Norway and that this may have contributed to the significant effect for prosocial framing. It is also interesting to note that the risk scenario had a greater impact on intentions to comply than the motivational emphasis. This may be due to the fact that both self-interest and prosocial frames have been demonstrated to enhance compliance with infection control measures [23, 26, 32, 35], while high versus low risk should have diverging effects on compliance. When conducting an exploratory analysis, we also found that prosocial motivation during the COVID-19 pandemic was associated with higher intentions to comply in the hypothetical scenario. It should be noted that the current motivation had a larger effect size for predicting future compliance than experimental factors of future risk and motivation factors. This goes to indicate that the variations in framing had a limited effect and may be confounded by other factors, whereas current behaviour is a good predictor of future behaviour.

There was no interaction effect of risk framing and motivational emphasis. This means that we did not find support for the expectation of higher compliance when the participants faced high risk and personal benefits were emphasized. One might expect that people are motivated to protect themselves when they assume that there is a high personal risk of infection. Isler and colleagues [27] found that people who perceived themselves to belong to an at-risk group were more persuaded to get vaccinated by information appealing to *personal* benefits, while people who considered themselves to be at low risk were more persuaded by prosocial framing. Contrary to this finding, our results suggest that prosocial motivation is more effective across levels of infection risk. To reconcile this, it may be that it is the severity of the consequence of getting infected and not the probability of infection that impacts the underlying motivation for compliance. In the current experiment, we operationalized the "high" and "low" risk scenario according to the probability of being infected rather than as the consequence of contracting the disease. Risk perception have been found to deviate from an expected utility calculation by being affected by decision frames [2, 3], but often both the probability and consequence of the equation is taken into consideration. Our choice to focus on probability rather than (or addition to) a focus on consequences may be defensible as the consequences of being infected may be somewhat similar (and somewhat uncertain) across our participants. Further, any differences (e.g., due to believing one is part of an at-risk group) should be randomized across our experiment groups. However, the focus on the probability side of risk may be part of the reason why we did not observe an interaction effect between risk and motivation. Future studies could manipulate the consequence instead of or in addition to the probability of contracting the disease to investigate if this aspect of perceived risk has different main effect or interaction with motivational framing.

## Limitations

The experiment is limited in that the outcome variable measured intention to comply with infection control measures rather than actual behaviour outcomes. Stated intentions may be influenced by social desirability bias such that participants respond in a manner that is viewed as favorable by others [36]. This may account for the effect of prosocial motivation if complying in order to limit an outbreak (i.e., prosocial motivation) is seen as a more virtuous than protecting oneself.

It should also be noted that the main effects explained little of the variation in intentions to comply and that prosocial motivational emphasis had a particularly modest impact. This is perhaps unsurprising as the experimental treatment constituted only subtle changes in the form of changing a few words in the phrasing of an otherwise similar hypothetical scenario. The effect sizes of the manipulated factors were smaller than what comparable research has previously shown [23, 26, 35]. Further, it has previously been shown than preregistered replications tend to have smaller effect sizes [37]. It should also be noted that the response distribution on the compliance item was skewed towards the high endpoint of compliance. This may indicate a ceiling effect, where there were less degrees of freedom to indicate the full potential of a possible effect. This may have contributed to the small effect sizes seen in the current results. Methodological limitations notwithstanding, the small magnitude of the effect may also indicate that the participant's intention to comply in a hypothetical scenario may be determined by a number of other factors in addition to the phrasing of risk and motivation. It has previously been argued [38] that change in behavioural intentions typically exaggerate the subsequent change in behaviour. It is therefore possible that the change in precautionary behaviour due to changing the risk scenario and motivation framing may be smaller still than what was observed in the experiment. Thus, the small effect sizes in our results may indicate that the practical impact of the communication framing on the level of compliance to infection control measures is limited.

It has been reported that trust in government institutions as well as intra-individual trust is unusually high in the Scandinavian countries, and that this may impact compliance [39, 40]. It could also be that the population's experience with the public health communication in past outbreaks has an impact on compliance with current infection control measures. Such factors should be taken into consideration when generalizing the results to other settings.

## Implications and future research

The response distribution for compliance was severely skewed towards the upper limits of high compliance. This happened despite taking measures to avoid this by phrasing the three response options that indicated highest compliance as more extreme (i.e., to emphasize that you would "always" follow the measures), based on similarly skewed results from a previous survey [24]. However, the far majority still indicated high intentions to comply across the experiment conditions. It thus appears that the more extreme phrasing did not cause greater variation in intentions to comply with control measures.

Our findings suggest that people are more willing to comply with infection control measures when they encounter a high-risk situation, and more willing still when the health authorities emphasized societal benefits for compliance (i.e., prosocial motivation). This suggest that in order to maximize compliance, information about the disease outbreak may benefit from emphasizing the public value of limiting infection spread and the risks that the disease represents. Although the effect sizes were small, the sample size and approach indicates robust results that to a population level may nevertheless translate to increased compliance for many people, and consequent quality-of-life improvements for those who avoid infection.

One may have expected that prosocial motivation is more relevant for low-risk scenarios, while it plays less of a role in high-risk scenarios. Interestingly, we found no interaction effects between risk scenario and motivational emphasis. Thus, it appears that the underlying motivation for complying with infection control measures operates independent of the infection risk facing individuals. Overall, the results indicated a high level of compliance across all experiment conditions. Future studies in this research project will complement the current experiment by measuring the association between types of motivation and compliance during pandemic phases with varying infection spread. The panel would benefit from targeted recruitment of people with lower levels of education and younger age to improve representativity for future studies.

## Supporting information

**S1 Dataset.**
(RDATA)

## Acknowledgments

Thanks to all the members of the PANDRISK research project for the discussions that contributed to the current data collection. We are particularly grateful to M. Knapstad and S. Øverland at the Public Institute of Health for feedback and direction on the framing of the experiment. Thanks to all the participants that took time to respond to the survey. The analyses in the current manuscript were publicly preregistered in advance of accessing the data (https://osf.io/ahfdn).

## Author Contributions

**Conceptualization:** Sebastian Bjørkheim, Bjørn Sætrevik.

**Data curation:** Sebastian Bjørkheim, Bjørn Sætrevik.

**Formal analysis:** Sebastian Bjørkheim.

**Funding acquisition:** Bjørn Sætrevik.

**Investigation:** Sebastian Bjørkheim, Bjørn Sætrevik.

**Methodology:** Sebastian Bjørkheim, Bjørn Sætrevik.

**Project administration:** Bjørn Sætrevik.

**Writing – original draft:** Sebastian Bjørkheim, Bjørn Sætrevik.

**Writing – review & editing:** Sebastian Bjørkheim, Bjørn Sætrevik.

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
