## [Decision Letter · Decision Letter 0]

12 May 2022

PONE-D-22-05740Manipulating risk of infection and appeal to public benefit increase compliance with infection control measures in a hypothetical pandemic scenarioPLOS ONE

Dear  Sebastian Brun Bjørkheim

Thank you for submitting your manuscript to PLOS ONE. After careful consideration, we feel that it has merit but does not fully meet PLOS ONE’s publication criteria as it currently stands. Therefore, we invite you to submit a revised version of the manuscript that addresses the points raised during the review process.

We look forward to receiving your revised manuscript.

Kind regards,

Gert G. Wagner

Academic Editor

PLOS ONE

Journal Requirements:

“The present study was conducted as a part of the PANDRISK research project funded by the Trond Mohn Foundation, project number TMS2020TMT08”

“The present study was conducted as a part of the PANDRISK research project funded by the Trond Mohn Foundation, project number TMS2020TMT08.

The funder had no role in study design, data collection and analysis, decision to publish, or preparation of the manuscript.”

3. Please upload a new copy of Figure 1 as the detail is not clear. Please follow the link for more information: " ext-link-type="uri" xlink:type="simple">https://blogs.plos.org/plos/2019/06/looking-good-tips-for-creating-your-plos-figures-graphics/"
https://blogs.plos.org/plos/2019/06/looking-good-tips-for-creating-your-plos-figures-graphics/

4. We note you have included a table to which you do not refer in the text of your manuscript. Please ensure that you refer to Table 1 in your text; if accepted, production will need this reference to link the reader to the Table.

Main issues:

Because the whole paper is about developing behavioral change measures (threatening and persuasion instead of information and communication), it should be written accordingly.

There is the question whether other factors besides education, age, gender that have an influence on compliance with protection measures (e.g., SES, race/ethnicity) have been considered. Can they be identified using the Norwegian Citizen Panel?

What is missing from the limitations is that the results are not transferable to other countries, as trust in national public health authorities and past risk communication (especially the COVID-19 risk communication) influence the trust of the population and thus have an impact on compliance with future infection control measures.

Further, there is no discussion in the limitations section on how the under-representation of certain groups (e.g., with low education) can be reduced in the future.

Reviewers' comments:

Reviewer's Responses to Questions

**Comments to the Author**

1. Is the manuscript technically sound, and do the data support the conclusions?

Reviewer #1: Yes

Reviewer #2: Partly

2. Has the statistical analysis been performed appropriately and rigorously? 

Reviewer #1: Yes

Reviewer #2: Yes

3. Have the authors made all data underlying the findings in their manuscript fully available?

Reviewer #1: Yes

Reviewer #2: Yes

4. Is the manuscript presented in an intelligible fashion and written in standard English?

Reviewer #1: Yes

Reviewer #2: Yes

5. Review Comments to the Author

Reviewer #1: Thank you for the opportunity to review the manuscript. The work is very interesting in terms of exploring different motivations for adherence to infection control measures during pandemics. However, it is unclear what recommendations for policy makers can be derived from the study results and to what extent best practices and ethical standards of effective (evidence-based) risk communication have been considered, aiming at empowering citizens in democratic societies to make informed decisions. For instance, evidence suggests that persuasive strategies threaten trustworthiness and credibility of the communicator and of vaccinations (e.g., Blastland et al. 2020; Kachurka et al. 2021; Volpp et al. 2021. Therefore, trust in public health authorities is a key factor in behaviour and compliance. We suggest, that different strategies of communication with the public as well as their chances of success and consequences should at least be discussed.

What is missing from the limitations is that the results are not transferable to other countries, as trust in national public health authorities and past risk communication (especially the COVID-19 risk communication) influence the trust of the population and thus have an impact on compliance with future infection control measures.

Further, there is no discussion in the limitations section on how the under-representation of certain groups (e.g., with low education) can be reduced in the future. There is also the question whether other factors besides education, age, gender that have an influence on compliance with protection measures (e.g., SES, race/ethnicity) have been considered. Can they be identified using the Norwegian Citizen Panel?

Please find below a few comments on some of the sections and phrases.

Abstract

In general, the methodology could be explained a bit more: sample, design, analyses.

It is not clear from the abstract what the main objective is and in how far the results of this study can inform future communication of health authorities. In 1.3.1 you wrote “The aim of the current study is to test causal relationships between risk of infection and type of motivation on complying with infection control measures, and to examine if these factors interact.” and in line 119 “Knowledge about the relative importance of self-interested and prosocial motives in different risk scenarios can inform health authorities about communication strategies during different phases of disease outbreak.” Maybe you can add this to your abstract.

Line 17 „Perceiving the situation as threatening and seeing public benefits to complying may increase the public’s motivation to comply.“

It is not clear what the conclusion for political actions could be? Should it be the aim of political communication be to raise fear (even if there is no reason for it or when there is a lot of uncertainty)? We view this statement critically in the light of evidence-based risk communication, which aims at transparent and balanced risk communication to enable informed decisions. If persuasion is the recommendation, it should be made more clear in the abstract.

Introduction

Line 38 “Although the probability of an event and the expected outcome can be established statistically, perceived risk has been shown to deviate from objective measures of risk (2,3). How threatening a pandemic is perceived to be, may be influenced by how public authorities communicate the risk scenario.”

The statistical probabilities of occurrence cannot be determined for all events, as also described in the paper by Mousavi Gigerenzer 2014 that you cite. Especially during the COVID-19 pandemic, a great deal of uncertainty existed at the beginning (and to some extent still exists). Therefore, a key factor influencing perceived risk is how risks and uncertainties are communicated by public authorities. However, evidence-based and user-centered risk communication not only includes transparent and balanced communication of probabilities, but also indicates the uncertainties, and takes into account the needs of different target groups). People better understand risks when, for instance, risk probabilities are communicated numerically instead of verbally (for example, “5 in 100 people will be infected” can be grasped more concretely than “The risk of infection is low or high”), since people tend to interpret verbal probability statements differently. Simple frequencies or per- centages (for example, 5 in 100 or 5%) are more comprehensible than probabilities or 1-in-x formats (for in- stance, 1 in 20). By communicating both numerator and denominator, it can be conveyed whether the risk is big or small. Moreover, when comparing risks, the denominator should be kept the same (for example always 100). Relative risk reductions are unclear (“The intervention reduced the number of infections by 20%”), in- stead, absolute risk reductions are recommended (”The intervention reduced the number of infections from 5 in 100 people without the treatment to 4 in 100 people with the treatment”) (e.g., Zipkin et al. 2014, Büchter et al. 2014). The latter illustrate the absolute size of a risk. If there is insufficient information to report numbers, the reason for this should be stated. Visual presentations can be a beneficial supplement or substitute for numerical or verbal information on risks. They can facilitate comprehension, especially for people with low numeracy or reading skills (e.g., McDowell et al. 2021).

Line 60 “If feeling threatened is essential for compliance, this would justify that health authorities emphasize personal and societal risks when promoting infection control measures.“

In the light of evidence-based risk communication, the benefits and harms of both perspectives (personal and societal risks) should be communicated if this is in the interest of the addressees.

Line 112 “How people perceive risk is related to their ability to control their level of exposure and it may thus be difficult to determine the impact of perceived risk on compliance from correlational data alone (25).”

How people perceive risks also depends on their risk literacy, which in turn is influenced, for example, by their statistical skills and the way in which risks, and uncertainties are communicated (e.g., Cokely et al. 2018). If risks are not correctly understood, risks can be overestimated or underestimated (e.g., Wegwarth et al. 2012), which in turn affects people's compliance for preventive measures. Transparent risk communication is therefore of central importance and should be the goal of public communication.

Line 129 “Earlier findings have indicated that perceived risks and different types of motivation as

important predictors of compliance”

Line 188 “This period has been referred to as the “second wave” of infection spread and a number of infection control measures were in place at the time (Ministry of Health Services, 2020). The rise in COVID-19 cases and the appropriateness of the infection control measures was prominent in the public debate during the data collection period.”

It would be helpful to readers to include details on average COVID-19 cases and on the infection control measures in the data collection period.

Discussion

Line 315 “Thus, policy efforts to prevent infection spread may focus on increasing either self-interested or prosocial motivation to comply“.

Or both, to address the different information needs and preferences as well as motivations of the target groups.

Reviewer #2: The authors submitted a scenario-based experimental study that varied infection risks. They renew a traditional finding: that perceived risk from a disease is associated with increased compliance to preventive measures.

I am suggesting to resubmit a modified manuscript.

Major reasons:

The manuscript in the current form does not convey a strong contribution. Compared to existing evidence: 1) The distinct association of risk perception (probability not always, severity yes) with reported preventive measures was shown before (e.g. COSMO studies in Germany and Spain; e.g. Rattay et al.). Now doing a study on risk-dependent intentions to comply (weaker endpoint) does not add much. 2) Intention to comply with public health measures in the light of societal benefits, while that directly depends on the perceived-benefit harm ratio of public health measures, does also add not much. Generally, the manuscript has to be settled more in the existing COVID literature or at least protection-motivation literature.

Meaningful results: Once you work with thousands of participants from quasi-representative surveys, for main effects significance testing is secondary to meaningful effects. In 3.2.1.: Main effect for high risk on compliance intention: partial eta square .03. What is the interpretation of not finding a larger effect? Extreme example 3.2.2.: with 2,521 degrees of freedom, can we speak about an effect with partial eta square .01? This is particularly important in the light of the interaction of motivation and risk: ones we acknowledge the relationship (higher risk -- higher personal risk perception -- more compliance with measures that protect me against this risk) -- why this was not shown [please adjust the y-axis range of Figure 1]? I do not have the answer. But this could be the key finding (e.g. confirming that probabilities are less important than severity in this context?). You should make use of the large dataset and explore potential confounders for that. Related to that - where are the descriptive statistics of risk perception and prosociality (manipulation checks)? The second paragraphs about small effects in limitations and in implications cannot replace that.

Conclusion, starting from the „starting point“: "the public must be informed about the infection risk and be motivated to comply with infection control measures". It lacks the main point: "the public must be informed about the benefits and harms of complying (and not complying) with infection control measures". As long as this is not clarified, "these findings can[not] inform communication strategies". If the whole paper is about developing behavioral change measures (threatening and persuasion instead of information and communication), it should be written accordingly. Related to that, goal of communication is to adjust risk perception of recipients in line with the objective risk (so, accurate risk perception), not to increase disease risk perception beyond, which can cause harm, similar to fear induction.

Minors:

- "Manipulating" seems unnecessary as the title expresses the finding

- Threatening and fear is not equal to risk perception - this needs to be clearly disentangled throughout the paper, e.g. p.3, l.40, when you talk in a paragraph about risk perception

- p.6 l.129: check grammar of the sentence please

- p.8 l.161-164: percentages of what (absolute, relative overrepresented)?

- Hypotheses and Design: what is "level of infection risk"? The stimuli could have produced a mixed understanding of perceived probability of infection (exposure risk in the low risk condition - "only a few people") and perceived severity of infecting (high risk: "serious outbreak of infection"). For another study, „many people“, „most of the people“, „nearly everybody“ would be better comparable. Was there information for the participants, explaining why following infection control advice stops the outbreak?

- p.10 l.210 and 211: check grammar of the stimulus sentences please

- consider removing sentences and expression, "something is significant", when you report statistics anyway and as you would not phrase non-significant differences as differences without comment.

- ethics: the study procedure did not deviate from the standard practice of the Norwegian Citizen Panel - with health data - complying in ongoing pandemic measures, experimental design? Could you attach the study consent form?

- what is the message of table 1 beyond the aggregate reporting? consider removing.

- 3.2 can be presumed by the reader, could be skipped

- p.17, l.382 "overstates"?

- some literature:

Rattay, P., Michalski, N., Domanska, O. M., Kaltwasser, A., De Bock, F., Wieler, L. H., Jordan, S. (2021). Differences in risk perception, knowledge and protective behaviour regarding COVID-19 by education level among women and men in Germany. Results from the COVID-19 Snapshot Monitoring (COSMO) study. Plos one, 16(5), e0251694.

Betsch, C., Wieler, L., Bosnjak, M., Ramharter, M., Stollorz, V., Omer, S., ... Schmid, P. (2020). Germany COVID-19 Snapshot MOnitoring (COSMO Germany): Monitoring knowledge, risk perceptions, preventive behaviours, and public trust in the current coronavirus outbreak in Germany. https://www.psycharchives.org/en/item/e5acdc65-77e9-4fd4-9cd2-bf6aa2dd5eba

Beca‐Martínez, M. T., Romay‐Barja, M., Falcón‐Romero, M., Rodríguez‐Blázquez, C., Benito‐Llanes, A., Forjaz, M. J. (2021). Compliance with the main preventive measures of COVID‐19 in Spain: The role of knowledge, attitudes, practices, and risk perception. Transboundary and Emerging Diseases.

Betsch, C., Korn, L., Burgard, T., Gaissmaier, W., Felgendreff, L., Eitze, S., ... Bosnjak, M. (2021). The four weeks before lockdown during the COVID-19 pandemic in Germany: a weekly serial cross-sectional survey on risk perceptions, knowledge, public trust and behaviour, 3 to 25 March 2020. Eurosurveillance, 26(42), 2001900.

Felix Rebitschek

I always sign my reviews

6. PLOS authors have the option to publish the peer review history of their article (what does this mean?). If published, this will include your full peer review and any attached files.

Reviewer #1: **Yes: **Christin Ellermann

Reviewer #2: **Yes: **Felix G. Rebitschek

---

## [Author Response · Author response to Decision Letter 0]

18 Aug 2022

Thank you for reviewing the manuscript. Please see the attached response letter "Response to reviewers".

---

## [Editor Report · Decision Letter 1]

22 Aug 2022

Manipulating risk of infection and appeal to public benefit increase compliance with infection control measures in a hypothetical pandemic scenario

PONE-D-22-05740R1

Dear Sebastian Brun Bjørkheim,

We’re pleased to inform you that your manuscript has been judged scientifically suitable for publication and will be formally accepted for publication once it meets all outstanding technical requirements.

Kind regards,

Gert G. Wagner

Academic Editor

PLOS ONE

---

## [Editor Report · Acceptance letter]

30 Aug 2022

PONE-D-22-05740R1 

Manipulating risk of infection and appeal to public benefit increase compliance with infection control measures in a hypothetical pandemic scenario 

Dear Dr. Bjørkheim:

I'm pleased to inform you that your manuscript has been deemed suitable for publication in PLOS ONE. Congratulations! Your manuscript is now with our production department. 

Kind regards, 

on behalf of

Professor Gert G. Wagner 

Academic Editor

PLOS ONE